# Raman Spectroscopy Combined with Malaria Protein for Early Capture and Recognition of Broad-Spectrum Circulating Tumor Cells

**DOI:** 10.3390/ijms241512072

**Published:** 2023-07-28

**Authors:** Xinning Liu, Yidan Zhang, Xunrong Li, Jian Xu, Chenyang Zhao, Jinbo Yang

**Affiliations:** 1Key Laboratory of Marine Drugs, Ministry of Education, Qingdao Marine Biomedical Research Institute, Ocean University of China, Qingdao 266071, China; lmm16890@163.com (X.L.);; 2Innovation Platform of Marine Drug Screening & Evaluation, Qingdao Marine Science and Technology Center, Qingdao 266100, China; 3Single-Cell Center, CAS Key Laboratory of Biofuels, Shandong Key Laboratory of Energy Genetics and Shandong Energy Institute, Qingdao Institute of Bioenergy and Bioprocess Technology, Chinese Academy of Sciences, Qingdao 266071, China; 4University of Chinese Academy of Sciences, Beijing 100000, China

**Keywords:** Raman spectroscopy, circulating tumor cell, malaria proteins, tumor marker, tumor metastasis

## Abstract

Early identification of tumors can significantly reduce the mortality rate. Circulating tumor cells (CTCs) are a type of tumor cell that detaches from the primary tumor and circulates through the bloodstream. Monitoring CTCs may allow the early identification of tumor progression. However, due to their rarity and heterogeneity, the enrichment and identification of CTCs is still challenging. Studies have shown that Raman spectroscopy could distinguish CTCs from metastatic cancer patients. VAR2CSA, a class of malaria proteins, has a strong broad-spectrum binding effect on various tumor cells and is a promising candidate biomarker for cancer detection. Here, recombinant malaria VAR2CSA proteins were synthesized, expressed, and purified. After confirming that various types of tumor cells can be isolated from blood by recombinant malaria VAR2CSA proteins, we further proved that the VAR2CSA combined with Raman spectroscopy could be used efficiently for tumor capture and type recognition using A549 cell lines spiked into the blood. This would allow the early screening and detection of a broad spectrum of CTCs. Finally, we synthesized and purified the malaria protein fusion antibody and confirmed its in vitro tumor-killing activity. Herein, this paper exploits the theoretical basis of a novel strategy to capture, recognize, and kill broad-spectrum types of CTCs from the peripheral blood.

## 1. Introduction

Malignant tumors have become one of the major public health problems that seriously threatens the health of the global population [1]. According to statistics, approximately eight million people die from cancer-related diseases each year, and researchers predict that this number will increase rapidly. Among them, more than 90% of cancer-caused deaths are related to metastatic disease [2]. Circulating tumor cells (CTCs) are unusual types of tumor cells that are shed from the primary tumor during metastasis and enter the bloodstream through blood vessels or the lymphatic system [3]. Such cells can spread to distant organs through blood circulation and proliferate in a suitable environment, which is the leading cause of tumor metastasis in patients. Early detection of CTCs in the blood is of great significance for patient survival assessment and therapeutic efficacy evaluation [4,5,6]. In recent years, more and more studies have been conducted on CTC identification, isolation, and characterization techniques [7,8,9,10]. The CellSearch system detects the epithelial-cell-derived CTC population by immunomagnetic enrichment. It is the only commercially available product in the world that is approved by the FDA for the clinical testing of CTCs [11,12]. However, due to phenotypic heterogeneity, CTCs in the blood are usually found with epithelial phenotypes, mesenchymal phenotypes, and mixed mesenchymal and epithelial phenotypes [13,14]. CTCs with mesenchymal phenotypes have reduced expression of epithelial markers such as EpCAM and CK keratin and, therefore, cannot be identified by the CellSearch-type method. Consequently, it is imperative to develop an effective technique for the enrichment of broad-spectrum CTCs.

Raman spectroscopy is a powerful and versatile analytical technique that provides valuable information about the structure, composition, and physical properties of a sample by measuring the energy changes in scattered photons. This method finds extensive applications in various fields, including (1) materials science, where Raman spectroscopy is used to identify the chemical compositions of different materials and detect impurities [15]; (2) environmental monitoring, where it helps to monitor contaminants in the environment, such as chemicals in water and gases in the atmosphere [16]; (3) energy-related research, where Raman spectroscopy plays a crucial role in studying the structure and properties of energy-related materials like fuel cells, energy storage materials, and solar cells [17]; (4) food analysis, where it aids in the detection of ingredients, additives, and nutrients in food, ensuring the quality and safety of the food supply [18]; (5) biomedical applications, where Raman spectroscopy is used to identify cells, tissues, and biomolecules, enabling rapid disease diagnosis and lesion monitoring [19]; and (6) drug discovery, where Raman spectroscopy can be utilized to analyze the chemical structure, purity, and stability of drugs, ensuring their quality [20]. The versatility and precision of Raman spectroscopy make it an indispensable tool in modern scientific research and various industries. In addition to its wide range of applications, as an analytical technique, Raman spectroscopy has several advantages, including (1) its non-destructive nature, as Raman spectroscopy preserves the integrity of the sample during testing. The sample remains unaltered, making it suitable for the analysis of valuable or irreplaceable samples; (2) its high sensitivity, as Raman spectroscopy can analyze trace samples at the nanogram level. Its high sensitivity enables the detection of even small quantities of substances; (3) its rapid analysis, as Raman spectroscopy provides spectral information about the sample within seconds. Its quick measurement time makes it a valuable tool for high-throughput analysis, real-time monitoring, and process control; and (4) the minimal sample preparation, as most samples can be directly measured by Raman spectroscopy without requiring extensive sample preparation or complex calibration procedures. This simplifies the analysis process and saves time. In summary, Raman spectroscopy’s non-destructive nature, high sensitivity, rapid analysis capability, and minimal sample preparation make it a crucial analytical tool in scientific research, industrial applications, biomedical fields, and beyond. Its versatility and user-friendly features contribute to its widespread adoption across diverse disciplines [21,22]. Currently, researchers are combining Raman spectroscopy with machine learning algorithms for sample identification and classification. Elumalai et al. used Raman spectroscopy to characterize urine from normal and oral cancer patients with an accuracy of up to 93.7%. Haka and colleagues demonstrated that Raman spectroscopy can distinguish among normal, benign, and malignant breast lesions in vitro with a specificity of 96% [23]. Studies have used EpCAM in combination with Raman spectroscopy to capture and identify CTCs in the peripheral blood [24,25]. However, previous studies still have obvious technical deficiencies and shortcomings, such as a low prediction sensitivity, simple classification modeling algorithms, and limited identification of tumor cell types.

The placenta of pregnant women expresses a specific type of chondroitin sulfate (CS), which binds tightly to the P. falciparum parasite and causes placental malaria [26,27]. Researchers have found that almost all solid tumor cells express this specific type of CS on the cell surface [28,29,30]. VAR2CSA, a malaria protein, has been shown to bind tightly to this particular type of CS and mediate the development of malaria [31]. Researchers found that VAR2CSA does not bind to normal cells but is positively correlated with various cancer cells [32,33,34]. Therefore, the sensitivity of using VAR2CSA to capture CTCs is much higher than that of traditional immunocapture methods. In this study, we expressed and purified VAR2CSA. The binding affinity between the malaria protein and various tumor cells was verified by flow cytometry. Based on the strong binding affinity of VAR2CSA to tumor cells, we explored the application of VAR2CSA in combination with Raman spectroscopy for tumor-type recognition. Different types of tumor cell lines were detected by Raman spectroscopy, and a recognition model was established. Subsequently, Raman spectroscopy combined with the malaria protein coupled with anti-biotin magnetic beads was able to capture and identify the tumor cells spiked into the blood. Our study demonstrates that, despite changes in the tumor type and heterogeneity, the model could successfully predict the captured tumor cell type. Further investigation of the in vitro tumor-killing activity of the VAR2CSA DBL2-Fc fusion antibody was also conducted. The relevant findings in this paper provide a theoretical basis, giving a better understanding of the characteristics of VAR2CSA, and have aided in the development of the clinical research and application of VAR2CSA coupled with Raman spectroscopy to capture, predict, and kill broad-spectrum CTCs.

## 2. Results

### 2.1. VAR2CSA DBL2 Can Efficiently Bind to Cancer Cells

To evaluate the activity of malaria proteins, VAR2CSA DBL1-2, DBL2, and SpyCatcher were expressed and purified as previously reported (Appendix A) [35]. To verify whether the purified proteins could be used to capture CTCs, we first examined the binding abilities of different types of tumor cells to malaria proteins by flow cytometry. As shown in Appendix A, the VAR2CSA DBL1-2 double-domain protein (250 nM) has a strong binding capacity with various tumor cells. Its binding activity to human-derived tumor cells (62% to 93%) is much higher than that to murine-derived tumor cells (12% to 38%), suggesting that the CS present on the surface of human-derived tumor cells may be very different from that of murine-derived tumor cells. The binding activity is similar to that reported previously [32]. Next, we selected A549 cells with a strong binding capacity for the following experiments. The binding strength of VAR2CSA DBL1-2 to mixed A549 cells and mouse leukocytes (in a ratio of 5:5) increased from 85% to 96% and 15% to 30% with a gradual increase in the protein concentration from 50 nM to 250 nM (Appendix A). Due to the large molecular weight and the difficulty of protein purification for VAR2CSA DBL1-2, we further investigated the binding ability of the VAR2CSA DBL2 single-domain protein with two cell types. The results show that the binding strength of VAR2CSA DBL2 to mouse leukocytes and tumor cells is significantly higher than that of VAR2CSA DBL1-2. At a low protein concentration (50 nM), VAR2CSA DBL2 bound to 96% of the A549 cells (Appendix A). With an increasing protein concentration, the binding of the single-domain protein to A549 cells increased from 96% to 99%, and binding to mouse leukocytes increased from 25% to 52%. Its binding strength to tumor cells was more than three times that of mouse leukocytes, and there were statistically significant differences in each concentration group (*p* < 0.01).

### 2.2. Taking Only PBMC Layer Cells Can Greatly Reduce the Non-Specific Binding of Malaria Proteins to Blood Cells

In order to improve the specificity of CTCs captured by VAR2CSA DBL2 in clinical applications, we further investigated the specific malaria protein binding subpopulation with leukocytes. The peripheral blood of mice mainly contains T cells, B cells, granulocytes, NK cells, monocytes, and dendritic cells. As shown in Appendix A, the various measured leukocyte proportions were the same as those reported previously [36]. Subsequently, our results show that the subpopulation of leukocytes bound to the malaria protein comprised 10.5% B cells (CD19^+^), 38% T cells (CD3^+^), and 45% monocytes as well as granulocytes (CD11b^+^) (Figure 1A,B). This suggests that malaria proteins mainly bind to T cells and granulocytes and very rarely to B cells, which are abundant in the blood. We then used density gradient centrifugation to determine the distribution of A549 cells in the mouse blood cells by absolute counts. Table 1 shows the relative numbers of different cell subsets in each layer, and Table 2 shows the absolute number of all cell subsets by the absolute number of fluorescent microspheres. The results show that the distribution of cells in the plasma and ficoll is very low, and CD11b^+^ cells are mainly distributed in the erythrocyte and monocyte layers, accounting for 57% and 28%, respectively. This indicates that most CD11b^+^ cell interference can be removed when only the PBMC layer is taken. In addition, we also found that, after density gradient centrifugation, A549 was mainly distributed in the PBMC layer, accounting for up to 96%. Therefore, taking only the PBMC layer cells can reduce the interference of irrelevant cells in the experiment and improve the detection limit of the investigation under the premise of losing tumor cells. To further simulate the actual situation of CTCs captured by the malaria protein in the peripheral blood, A549 cells were mixed with normal human leukocytes, and the PBMC layer was collected after density gradient centrifugation. With an increase in the protein concentration, the binding of VAR2CSA DBL2 to both A549 and human leukocyte PBMC rose from 64% to 82% and from 4% to 20%, respectively. This result indicates that the binding strength of PBMC to the malaria protein decreased after density gradient centrifugation and was statistically significantly lower than that of whole blood (*p* < 0.01) (Figure 1C,D).

### 2.3. Malaria Protein Is a More Broad-Spectrum Tumor Surface Marker than EpCAM

The primary tumor surface marker that is currently commercially available for CTC detection is the epithelial cell marker EpCAM. However, due to the EMT transformation that occurs when tumors enter the peripheral blood, traditional surface markers may miss a fraction of tumor cells of mesenchymal origin. In this section, we compare the binding ability of the malaria protein and EpCAM to different types of tumor cells. As shown in Figure 1F, we first identified mesenchymal-derived cells MCF7 and epithelial-derived cells HCT116. Further, we verified the binding strength of the malaria protein and EpCAM to these two cells by flow cytometry. As shown in Figure 1H, the malaria protein was found to have a high binding ability to both HCT116 and MCF7 cells with a binding rate of approximately 90%. These results indicate that the binding of malaria protein to tumor cells is not affected by the cell type. However, the expression of EpCAM was significantly different in the two cell types, with 100% binding to the epithelial type HCT116 and only 7% binding to the mesenchymal type MCF7. This indicates that EpCAM, a marker of epithelial cells, cannot bind to mesenchymal-type tumor cells. We further compared the binding of VAR2CSA DBL2 and EpCAM to MCF7 mesenchymal cells. As shown in Figure 1G,I, 89% of MCF7 cells (Vimentin^+^) can bind to VAR2CSA DBL2 (Q2/Q2+Q3), whereas only 15.8% of MCF7 cells (Vimentin^+^) can bind to EpCAM (Q2/Q2+Q3). The expression of VAR2CSA DBL2 in MCF7 is statistically significantly higher than EpCAM (*p* < 0.01). Therefore, malaria proteins are advantageous over EpCAM as surface markers to capture CTCs.

### 2.4. Raman Spectroscopy Can Predict Cell Type Successfully

Clinically, early monitoring of CTCs may reflect tumor progression, but detection alone appears to be insufficient. Only identifying early tumor types can lead to more personalized treatment decisions, curbing tumor progression from the source and improving patient survival. Previous results have confirmed that the malaria protein is a broader-spectrum tumor marker than EpCAM. Therefore, we investigated the differentiation and identification of CTCs and leukocytes in patients’ peripheral blood by Raman spectroscopy using VAR2CSA as a tumor marker. We first constructed spectral models of eight different types of cell lines by Raman spectroscopy, including six adherent tumor cells (A549, DU145, HCT116, HepG2, MCF7, Hela), one suspended tumor cell (HL60), and one normal cell (BJ). The content and components of substances within the different cell types can be reflected in the spectrograms (Figure 2A). We established the classification model through six common algorithms: PLS-DA, SVM, K-NN, RF, LDA, and XGB (Table 3). The average prediction accuracy for the eight types of cells had an average value of 99%; the prediction success rate for the eight cell types was 100%, indicating that the established cell models perform well. Having found that Raman spectroscopy can predict the same batch of samples very well, we tried to distinguish the cells from different batches using the RF algorithm. It was found that the prediction accuracy was reduced to varying degrees when predicting different batches of cells. Only A549, Hela, and HL60 could be successfully predicted (Table 4). This result suggests that the generalization ability of the model needs further improvement. Meanwhile, when the prediction model of HL60 and DU145 cells was established using eight tested cells, we found that the prediction accuracy of HL60 cells was 60%, higher than other test cells. It is speculated that HL60 cells are suspended cells that do not require digestion, have fewer pre-processing steps, and retain more cell characteristics.

### 2.5. Raman Spectroscopy Can Identify Tumor Cell Types Captured by Malaria Proteins

After establishing and confirming the performance of the Raman spectroscopy prediction model, we used Raman spectroscopy to compare laboratory-cultured A549 cells and malaria-protein-captured A549 cells to the established cell model using six algorithms for cell-type identification. First, the flow cytometry analysis revealed that the malaria protein capture system had a high binding capacity for tumor cells (Figure 2B). The cell counting results confirmed that the malaria protein capture system can successfully capture tumor cells, and the capture rate of A549 was statistically significantly higher than that of leucocytes (*p* < 0.05) (Figure 2C). As shown in Table 5 and Table 6, both laboratory-cultured A549 cells and malaria-protein-captured A549 cells can be successfully predicated as A549 cells using six analysis algorithms. Among them, laboratory-cultured A549 cells can be predicted correctly by the PLS-DA algorithm, while malaria-protein-captured A549 cells can be predicted correctly by the SVM, PLS-DA, and XGB algorithms. In addition, laboratory-cultured A549 cells were predicted to have a higher probability of becoming HepG2 cells. We further compared the similarity between laboratory-cultured A549 cells and malaria-protein-captured A549 cells in the same batch. The predicted results for the six analysis algorithms are shown in Table 7. The A549 cells were successfully predicted by the SVM, RF, and XGB algorithms with probability of prediction rates of 74.47%, 46.81%, and 89.36%, respectively. Therefore, it is concluded that the tumor cells captured by the malaria protein still have similar characteristic spectra to those cultured in the laboratory, confirming the feasibility of the combination of malaria protein and Raman spectroscopy to capture, clarify, and identify tumor cells.

### 2.6. Raman Spectroscopy Can Identify Tumor Cells Captured by Malaria Proteins after Their Mixture with Peripheral Blood

After confirming that the malaria-protein-captured tumor cells still had a characteristic spectrum similar to laboratory-cultured cells, we mixed laboratory-cultured A549 with mouse leukocytes to test whether Raman spectroscopy can clinically type-identify malaria-protein-captured CTCs. We captured PBMC layer cells with the malaria protein after density gradient centrifugation. The results show that the cell types captured by malaria protein can be successfully predicted as A549 cells by the PLS-DA algorithm with a prediction probability of 46.67% (Table 8). These results indicate that the malaria-protein-captured cells can be accurately identified by Raman spectroscopy after mixing tumor cells cultured in the laboratory with leukocytes. However, the statistical analysis showed no significant difference with other cell types, and there was still some confusion, so the experimental scheme needs further optimization. This conclusion provides an empirical basis for the future clinical application of Raman spectroscopy combined with the malaria protein to detect tumor cells.

### 2.7. VAR2CSA DBL2-Fc Fusion Antibody Has a Strong Binding Ability with Tumor Cells

To assess the activity of the VAR2CSA DBL2-Fc fusion antibody, the fusion antibody was expressed and purified using Protein G-affinity chromatography. The expression of the VAR2CSA DBL2-Fc fusion antibody was analyzed by performing SDS-PAGE and Western blotting (Appendix A). The purified VAR2CSA DBL2-Fc fusion antibody showed a single, distinct band on an SDS-PAGE gel, indicating a molecular weight of about 130 kDa compared to the standard molecular weight markers.. To verify the purity of the final product, gel filtration chromatography was conducted, and the results are presented in Appendix A. The chromatogram of the purified protein showed a single peak, indicating that the protein exists as a stable monomer.

After expressing and purifying the VAR2CSA DBL2-Fc fusion antibody, we first confirmed its binding strength to the tumor by flow cytometry and immunofluorescence. As shown in Figure 2D, the flow cytometry results showed that both HCT116 and A549 cells could bind to the VAR2CSA DBL2-Fc fusion antibody, and A549 cells could bind to the VAR2CSA DBL2-Fc fusion antibody to a high degree in a concentration-dependent manner, with a binding ratio above 90%, and saturation at 400 nM (Figure 2E). The binding ability of the VAR2CSA DBL2-Fc fusion antibody to tumor cells was similar to that of the VAR2CSA DBL2 protein without an Fc fragment, indicating that the addition of an Fc fragment did not affect the binding ability of the protein to tumor cells. In addition, we visualized the binding effect by immunofluorescence, and the results again showed a high binding affinity between the fusion antibody and A549 cells (Figure 2F).

### 2.8. VAR2CSA DBL2-Fc Fusion Antibody Enhances Macrophage Antitumor Activity In Vitro

Antibody-dependent cellular cytotoxicity (ADCC) is a mechanism of monoclonal antibody therapy. The Fc domain of the antibody binds to the activating FcγR of the immune cells (NK cells, monocytes, macrophages, and granulocytes) and triggers the killing of a target cell. After confirming the high binding affinity between the VAR2CSA DBL2-Fc fusion antibody and A549 cells, we further tested the ADCC effect of the fusion antibody. A549 cells were mixed with mouse spleen cells and incubated with different concentrations of the fusion antibody for 48 h. As shown in Figure 3A, with a fusion antibody concentration of 0 or 0.5 μΜ, the cell morphology of A549 was normal, and almost no cell death was observed. When the concentration was increased to 2.5 μΜ, some cell necrosis was observed (the part pointed out by the red arrow), which means that the fusion antibody has a killing effect on the tumor cells. LDH is an enzyme located in the cytoplasm. When the cell membrane is damaged, the cells will release LDH into the culture medium, and measuring the level of LDH in the culture medium can determine the number of dead cells. Therefore, in this part of the experiment, we evaluated the ability of the VAR2CSA DBL2-Fc fusion antibody to induce ADCC by measuring the amount of LDH released from the cells. The results are shown in Figure 3B. A549 cells were used as target cells, and freshly isolated murine spleen cells and NK cells were used as effector cells. A549 and mouse spleen/NK cells were co-incubated with the fusion antibody for 6 h. The LDH assay was performed by taking the cell supernatant. It was found that the killing effect of the antibody on the target cells increased from 5% to 30% as the concentration increased from 0 to 2.4 μΜ. This killing effect was concentration-dependent. The intensity of the killing effect on target cells was similar for spleen cells and NK cells (*p* < 0.01). These results suggest that the VAR2CSA DBL2-Fc fusion antibody has an ADCC effect and can promote the killing effect of immune cells on tumor cells.

### 2.9. VAR2CSA DBL2-Fc Fusion Antibody Induces Macrophage-Dependent Phagocytosis in A549

In addition to the ADCC effect, ADCP is considered one of the primary mechanisms for the killing effect of antibody drugs on tumor cells and the binding of target cells with FcγR on the antibody–drug complex induces phagocytosis. We mixed A549 cells and mouse macrophages, labeled them, and incubated them with different concentrations of fusion antibody for 3 h. The phagocytosis rates were examined by flow cytometry. As shown in Figure 3C, in the A549 macrophage co-culture system, compared with the untreated group, the phagocytosis efficiency of macrophages in the co-culture system was improved to a certain extent by the fusion antibody, and the phagocytosis efficiency was concentration-dependent. The phagocytosis of macrophages was twice that of the non-treated group at a fusion antibody concentration of 2.4 μΜ. These results indicate that the purified VAR2CSA DBL2-Fc fusion antibody has an ADCP effect and can promote the phagocytosis of macrophages to target cells (A549) in the co-culture system (*p* < 0.05).

## 3. Discussion

Ninety percent of deaths in cancer patients are caused by tumor metastasis. Studies have shown that tumor metastasis can occur in the early stage of tumor onset. Early diagnosis of cancer is the primary strategy to improve patient survival. CTCs are present in peripheral blood several years before clinical diagnosis and are considered to be the seed of tumor transmission [37,38]. During the clinical treatment of cancer, monitoring the change in the number of CTCs is of great practical significance for assessing treatment prognosis and recurrence in patients. For instance, a decrease in the number of CTCs after the treatment indicates that the treatment is effective. Conversely, an increasing number of CTCs suggests a tumor recurrence, and a remaining high number of CTCs indicates a poor prognosis. This means that a new treatment strategy should be considered. In addition, dynamic CTC monitoring can quickly determine the follow-up treatment and provide a good reference for clinical medication. Traditional chemotherapy methods can only be evaluated after about 1 month of treatment, while changes in CTCs levels can be observed after 1 week of administration. In addition, longitudinal monitoring of changes in the number of CTCs in the same patient at different time points is essential. Studies have shown that the incidence of lung cancer in patients with pre-existing chronic obstructive pulmonary disease (COPD) is three times higher than in the general population. Regular CTCs and CT screening of patients with COPD revealed that the presence of tumors detected by CT could be 1–4 years later than the CTCs, suggesting that monitoring of the CTC levels could be used for the early detection of tumors [39]. Similarly, most liver cancer patients progress from hepatitis B to cirrhosis and eventually to liver cancer. Early monitoring of CTC counts in patients with cancer tendencies can provide an early warning of the occurrence of liver cancer [40]. Therefore, the clinical significance of CTC detection is beyond doubt.

Currently, CTCs research faces two main problems: cell rarity and heterogeneity. The CellSearch system, as the gold standard for CTC detection, utilizes the epithelial cell surface marker EpCAM for CTC capture, resulting in false positives due to the presence of many epithelial-type cells in the peripheral blood and false negatives because of the presence of mesenchymal-derived CTCs, which are downregulated or non-express epithelial cell surface markers. Presently, the capture principle of CTCs of non-epithelial origin, such as melanoma cells, is to distinguish generally based on the size difference between tumor cells and normal cells and label them by antigens specifically expressed by such tumor cells. This method is expensive and requires multiple antibody markers to determine the tumor type. Therefore, finding a new broad-spectrum approach to capture and kill these rare and vital cells could significantly improve cancer-related diagnosis, personalize treatment strategies, and reduce healthcare costs.

Herein, we reported a Raman spectroscopy-based microfluidic immunoassay for identifying CTC types. A previous model study detected tumor cell types by Raman spectroscopy using a model conducted by 2–3 types of cells [41,42,43]. In this research, we first provided an application of the predicted broad spectrum of CTCs when mixed with leukocytes using a classification modeling algorithm conducted by eight different kinds of cells. Raman spectroscopy could accurately distinguish multiple tumor cells in the same batch and a fraction of different batches of tumor cells. Additionally, we selected a recombinant malaria VAR2CSA protein as a biomarker for tumor capture, which was found to be expressed in more than 90% of the mesenchymal- and epithelial-derived cells. However, EpCAM, a well-known CTC biomarker, underwent 100% binding to the epithelial type and only 7% binding to the mesenchymal type (Figure 1H). Judging from these results, our research is superior to the previous models with regard to predicting the tumor cell type.

Antibody–drug conjugate drugs (ADCs) are now the fastest-growing drug class in oncology. They internalize and provide a means for delivering cytotoxic drugs into target cells. ADCs are composed of three key elements: a small molecule drug that selectively binds to an antigen on the tumor cell surface, the best feature of mAb, and a cleavable or non-cleavable linker. They have manageable safety issues by creating a single moiety that is highly specific and cytotoxic. Eleven different ADCs have been approved by the FDA for clinical use, and more than 100 ADCs are currently under evaluation in clinical trials worldwide [44,45]. As a seed of tumor metastasis, the direct killing of CTCs can inhibit metastasis and slow down tumor progression. Malaria proteins conjugated with toxic small molecules have been previously reported for tumor killing. But, because of the non-specific binding of malarial proteins, this approach may have strong side effects [28,32]. Hence, we designed the malarial protein–Fc fragment fusion antibody to directly bind to CTCs, induce an immune response, and kill CTCs through ADCC and ADCP by targeting the recognition of tumor cells by the malarial protein. In this paper, the purified malarial protein VAR2CSA was synthesized and expressed by a prokaryotic expression system. Based on determining the binding affinity of the malarial protein to tumor cells and leukocytes, the application of the malarial protein combined with Raman spectroscopy to capture and identify CTCs was explored. At the same time, the malaria protein fusion antibody was purified and synthesized to discuss its tumor-killing effect in a cell model. The current findings provide the theoretical grounds for application development and further research on malarial protein capture in CTCs.

## 4. Materials and Methods

### 4.1. Cell Culture and Antibodies

Human HCT116, A549, HepG2, Hela, DU145, MCF7, HL60, and BJ cell lines and ouse 4T1, B16, MC38, and LLC cell lines were obtained from the Shanghai cell line bank. 293 cell was obtained from the Union-Biotech company. HCT116 was grown in McCoy’s 5A (Modified) medium supplemented with 10% fetal bovine serum. A549, HepG2, MCF7, Hela, B16, LLC, DU145, MC38, and BJ cells were grown in Dulbecco’s Modified Eagle’s Medium (DMEM) with 10% fetal bovine serum, and 4T1 and HL60 cells were grown in RPMI-1640 medium with 10% fetal bovine serum. 293 cell was grown in UP1000 medium. Anti-Vimentin (5741S), anti-Vimentin-PE (12020S), anti-His Tag-FITC (14930), and anti-biotin (5597S) antibodies were purchased from Cell Signaling Technology (Danvers, MA, USA). Anti-CD11b-PE-Cy7 (1956821), anti-CD19-APC (11988589), and anti-IgG-AF488 secondary (A-10680) antibodies were purchased from Invitrogen (Waltham, MA, USA). Anti-CD3-PE (100205), anti-CD45-APC (368512), and anti-CD49b-PE (108908) antibodies were purchased from Biolegend (San Diego, CA, USA). The anti-CD45-PE-Cy5.5 (550994) antibody was purchased from BD (Franklin Lakes, NJ, USA). The anti-IgG-AF488 secondary (abs20013) antibody was purchased from Absin (Shanghai, China).

### 4.2. Cloning, Expression, Purification, and Molecular Weight Determination of VAR2CSA DBL1-2, VAR2CSA DBL2, and SpyCatcher

The target genes of VAR2CSA DBL1-2 and VAR2CSA DBL2 were synthesized and subcloned into the XhoI and NcoI restriction sites in the pET-28a (+) expression vector and transformed into *E. coli* DH5α cells by Shanghai Personalbio Technology Co., Ltd. (Shanghai, China). The recombinant plasmids were subsequently transformed into host cells to obtain the VAR2CSA DBL1-2 and VAR2CSA DBL2 proteins in Shuffle T7-K12 cells (BC208-01, Biomed, Beijing, China). The recombinant plasmid of SpyCatcher was obtained from Addgene and transformed into *E. coli* BL21 (DE3) (EC1002, Weidi, Shanghai, China) cells. Afterwards, a single colony was inoculated into an LB medium containing the corresponding antibiotics. The culture was then incubated on a rotary shaker at 37 °C until it reached an optical density at 600 nm (OD600) of approximately 0.8. Then, IPTG was added to the culture to obtain a final concentration of 200 µM, inducing protein expression. The bacterial culture was further incubated for 18 h at 16 °C. Following the incubation, the cells were harvested and subjected to sonication in a buffer solution containing 25 mM Tris–HCl (pH 7.5), 150 mM sodium chloride, 20 mM imidazole, 0.2 mg/mL lysozyme, 1 mM MgCl_2_, 1 mM PMSF, and 0.05% Triton X-100. Cellular debris was then removed by centrifugation at 12,000 g for 45 min at 4 °C. The resulting supernatants were collected and loaded onto a Ni Sepharose column (SA035005, Smart Lifesciences, Changzhou, China). The column was extensively washed with a washing buffer [25 mM Tris–HCl (pH 7.5), 150 mM NaCl, and 20 mM imidazole] to remove non-specifically bound proteins. Next, the target proteins were eluted using a Tris–HCl buffer with a gradient of 40 to 200 mM imidazole. Subsequently, the eluted proteins were desalted using a prepacked column containing 25 mM Tris–HCl (pH 7.5) and 150 mM NaCl to remove imidazole and other small molecules. The final protein was stored at −80 °C without any further modifications. To confirm the purity of VAR2CSA DBL2, VAR2CSA DBL1-2, and SpyCatcher, sodium dodecyl sulfate-polyacrylamide gel electrophoresis (SDS-PAGE) was performed. The proteins were separated on polyacrylamide gels using the method described by Laemmli (1970) and then stained with Coomassie Brilliant Blue R-250 (KGP1001, KeyGen, Nanjing, China). The molecular weights of the proteins were determined by comparing their migration rates with those of standard prestained markers with known molecular weights ranging from 15 to 180 kDa.

### 4.3. Gel Filtration Chromatography

The purified VAR2CSA DBL1-2, VAR2CSA DBL2, SpyCatcher protein, and VAR2CSA DBL2-Fc fusion antibody underwent gel filtration chromatography analysis using a Superdex 75 column with a flow rate of 0.5 mL/min. The sample was prepared in PBS buffer, and approximately 100 µL of the prepared sample was injected into the column. The detector (Qite, Shanghai, China) of the chromatography system was set at 280 nm, and the total running time for the analysis was 100 min. During the analysis, the retention time for each peak observed in the chromatogram was carefully noted.

### 4.4. Flow Cytometry

For the detection of VAR2CSA and the tumor cell binding ability, tumor cells were grown to 70–80% confluence in appropriate growth media and harvested in an EDTA detachment solution. A serious concentration of VAR2CSA in PBS containing 2% fetal bovine serum (FBS) was added to the tumor cells and incubated at 4 °C for 30 min. The binding ability was analyzed after incubating antibodies for 30 min at 4 °C in the dark. To detect the binding ability of VAR2CSA with leukocytes and tumor cells, male C57 mice aged 8–10 weeks were killed by the inhalation of carbon dioxide. Blood was collected from the hearts of the mice and placed in an anticoagulant tube. Peripheral blood mononuclear cells (PBMCs) were isolated using a density gradient medium. Tumor cells were processed as described above and mixed with leukocytes. VAR2CSA was diluted to the indicated concentration, added to the mixed cells, and incubated at room temperature for 30 min. Binding was analyzed after the incubation with antibodies for 30 min at 4 °C in the dark. The stained cells were analyzed using FACSAria (BD Biosciences, Franklin Lakes, NJ, USA) and FlowJo software version 10.

### 4.5. Capturing Cancer Cells by VAR2CSA-Coated Anti-Biotin Beads

SpyCatcher was biotinylated with NHS-biotin (Smart-Lifesciences, Changzhou, China) and detected by Western blot. VAR2CSA DBL2 and the biotinylated SpyCatcher were mixed in a 1:1 ratio and left for 2 h at room temperature to form a covalent isopeptide bond via the SpyTag–SpyCatcher interaction, as previously described [46]. The biotinylated VAR2CSA DBL2 was co-cultured with tumor cells at room temperature for 2 h, and anti-biotin bead (Miltenyi, Berlin, Germany) solution was added to the cell suspension at 4 °C for 15 min to capture the VAR2CSA DBL2 binding cells.

### 4.6. Raman Microspectroscopy

All Raman measurements were performed with a modified Raman spectrometer (Horiba, Oberursel, Germany). The cells were harvested after digestion. Cell pellets were fixed with 1% paraformaldehyde for 20 min at room temperature, centrifuged at 100× *g* for 4 min, and the supernatant was discarded. The fixed cells were resuspended with PBS on CaF_2_ slides for Raman spectroscopy testing. The Raman spectra of each sample were measured and analyzed using 532 nm laser excitation. The output power was 100 mW, the objective lens was 60×, and the wave number range of the Raman spectra recorded was 400–3200 cm^−1^. The samples were fixed on the three-dimensional XYZ platform. One cell was randomly selected under the microscope to be photographed. A total of 50–70 cell spectra were acquired for each cell type with an exposure time of 5 s per cell. For the data analysis, the anomalous spectra with signal-to-noise ratios of less than 10 and distortion were removed, the cosmic rays were eliminated from the remaining data, and the fluorescence background was removed to obtain a more accurate spectrum. The spectrum was pre-processed by Labspec5 software using the first-order derivative with the Savitzky–Golay smoothing method with a moving window width of nine and a polynomial number of three. All original spectra were normalized to the internal standard of the highest Raman peak area. SIMCA-P software was used to perform the principal component analysis with a downscaled projection for preliminary data visualization in the low-dimensional space.

### 4.7. Construction, Expression, and Purification of the VAR2CSA DBL2-Fc Fusion Antibody

The target genes of VAR2CSA DBL2 and mouse IgG 2a were synthesized and subcloned into the pcDNA3.1 expression vector and transformed into *E. coli* DH5α cells by Shanghai Personalbio Technology Co., Ltd. (Shanghai, China). IgG 2a was fused at the C-terminus with a 6× His-tag. The recombinant plasmid was then transformed to obtain the VAR2CSA DBL2-Fc fusion antibody in 293T cells. The media was harvested, and the protein was purified by protein G affinity chromatography (Smart-Lifesciences, Changzhou, China).

### 4.8. ADCP

Male C57 mice, aged 8–10 weeks, were injected intraperitoneally with 1 mL thioglycollate medium and sacrificed by carbon dioxide inhalation 3 days after injection. Peritoneal macrophages were harvested as previously described [47] and stained with violet CellTracker (C10094, Thermo, Waltham, MA, USA). A549 cells were harvested and stained with green CellTracker (C2102, Thermo, Waltham, MA, USA). The mixed peritoneal macrophages and A549 stained cells were treated with a series of concentrations of the VAR2CSA DBL2-Fc fusion antibody and plated in 24-well plates without TC treatment and then co-cultured in an incubator at 37 °C for 3 h. After co-cultivation, the cells were digested, washed twice with PBS, suspended in PBS, and analyzed by flow cytometry.

### 4.9. ADCC

The ADCC assay was conducted using the lactate dehydrogenase assay kit (Jiancheng, China), following the manufacturer’s instructions. Briefly, NK cells were prepared from the mouse spleen using an NK cell isolation kit (Miltenyi, Berlin, Germany). A549 cells (target cells) pretreated with the VAR2CSA DBL2-Fc fusion antibody (0.5, 2.5 μM) were incubated with mouse NK cells or spleen cells (effector cells) at a 1:5 effector:target ratio for 6 h at 37 °C. After incubation, the plates were centrifuged at 300 g for 5 min, and the supernatant was transferred to a 96-well plate to determine the amount of LDH released. The percentage of specific cell lysis was calculated as follows: (experimental release—background release/maximum release—background release) × 100%. Measurement of target cells treated with 1% Triton X-100 alone was used as high-release control.

### 4.10. Immunofluorescence Assays

A549 cells were seeded in six-well plates and then cultured at 37 °C for 12 h. After treatment with the VAR2CSA DBL2-Fc fusion antibody for 2 h, the cells were fixed with 4% paraformaldehyde for 20 min at room temperature, permeabilized with 0.1% Triton X-100 for 20 min, and washed three times with PBS. Cells were then incubated with the goat anti-mouse IgG-Alexa Fluor 488 antibody and 5 μM 4’,6-diamidino-2-phenylindole (DAPI) (H-1200-NB, Vector Laboratories, San Francisco, CA, USA) for 3 h. Plates were washed three times with PBS. Images were captured and analyzed using a Zeiss Primovert (Zeiss, Oberkochen, Germany).

### 4.11. Statistical Analysis

Graphing, data distribution, and statistical analysis were performed using QtiPlot. T-tests were utilized to investigate significant differences between the indicated groups. The data are presented as the mean ± SD of three independent experiments. Statistically significant results are denoted by asterisks, as follows: * *p* < 0.05, ** *p* < 0.01, *** *p* < 0.001, where *p* < 0.05 was considered statistically significant.

## 5. Conclusions

In conclusion, this article demonstrates the therapeutic efficacy of Raman spectroscopy combined with the malaria protein to capture, detect, and kill tumor cell lines spiked into the blood. We provided a basic experimental procedure for the early detection of tumors. Further investigations, such as multi-batch processes with large sample sizes for modeling and deep learning to differentiate fine-grained differences with more accurate data analyses, are needed to improve the accuracy of the experiment.

## Figures and Tables

**Figure 1 ijms-24-12072-f001:**
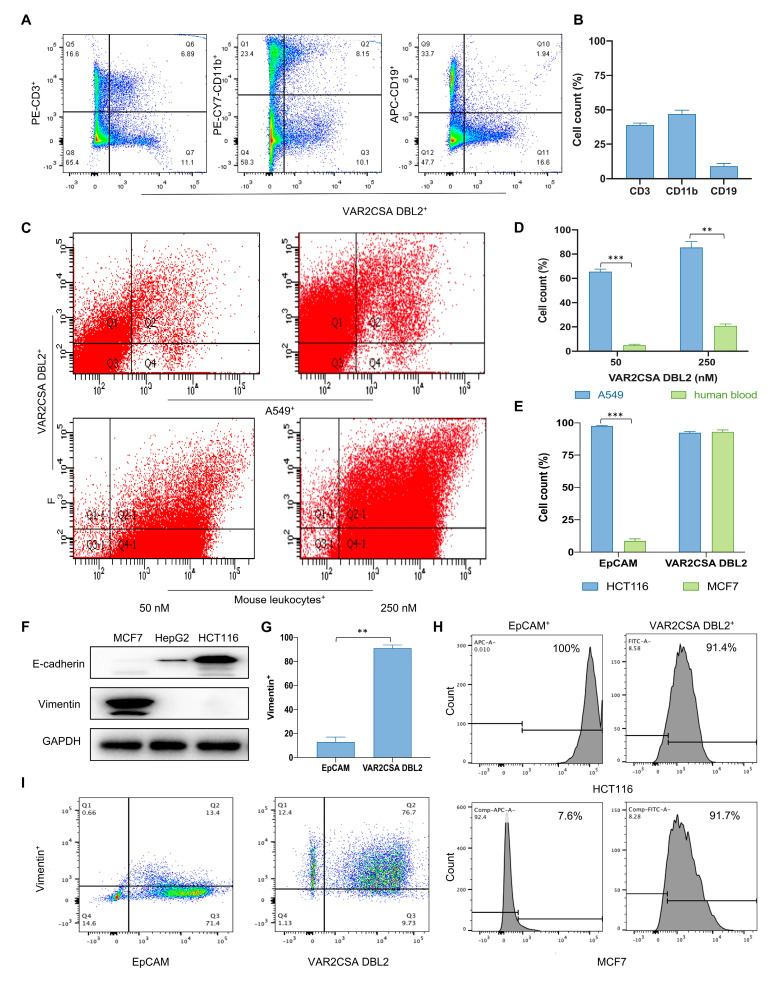
The malaria protein undergoes less non-specific binding to PBMC and has a broader binding spectrum than EpCAM. (**A**). The binding fraction of the malaria protein VAR2CSA DBL2 to leukocytes was determined by flow cytometry. Blood was obtained from mouse hearts. After lysis of the erythrocytes, 250 nM of the VAR2CSA DBL2 protein was added to the cells and incubated for 30 min at room temperature. T cells, granulocytes or monocytes, B cells, and the VAR2CSA protein were labeled with anti-CD3-PE, anti-CD11b-PE-Cy7, anti-CD19-APC, and anti-His tag-AF488 antibodies, respectively, and incubated at 4 °C for 30 min in the dark. (**B**) The relative quantification of (**A**) was performed. The binding affinities of VAR2CSA DBL2 to T cells, granulocytes or monocytes, and B cells were calculated as Q6/Q6+Q7, Q2/Q2+Q3, and Q10/Q10+Q11, respectively. (**C**) The binding affinity of different concentrations of VAR2CSA DBL2 for A549 mixed with human leukocytes was determined by flow cytometry. The A549 cells were digested by trypsin. Human blood was collected from human veins, and PBMC layers were obtained by density gradient centrifugation. The two kinds of cells were mixed in a 5:5 ratio. VAR2CSA DBL2 protein diluted to the appropriate concentration was added to the cells and incubated for 30 min at room temperature. The VAR2CSA protein (FITC) was labeled with the anti-His tag AF488 antibody, human leukocytes were labeled with the anti-CD45-APC antibody, and A549 cells were labeled with DAPI and incubated at 4 °C for 30 min in the dark. (**D**) The relative quantification of (**C**) was performed. The binding affinities of VAR2CSA DBL2 to A549 cells and human leukocytes were calculated as Q2/Q2+Q4 and Q2-1/Q2-1+Q4-1, respectively. ** *p* < 0.01, *** *p* < 0.001 when compared to the human leukocyte group. (**F**) The detection of different tumor cell sources. Different types of tumor cells were lysed, and the total protein content was extracted to determine the origin (epithelial or mesenchymal) of the tumor cells by Western blot. (**G**) The relative quantification of (**I**) was performed. ** *p* < 0.01 when compared to the EpCAM expressed in the MCF7 group. (**H**) Detection of the binding activity of VAR2CSA DBL2 and EpCAM to different origins of tumor cells by flow cytometry. HCT116 and MCF7 cells were digested by trypsin, an anti-His tag-AF488 antibody was added to label the VAR2CSA protein (FITC), and an anti-EpCAM-APC antibody was added to label EpCAM. (**E**) The relative quantification of (**H**) was performed. *** *p* < 0.001 when compared to the EpCAM expressed in the MCF7 group. (**I**) Detection of the binding activity of VAR2CSA DBL2 and EpCAM to mesenchymal cells MCF7 by flow cytometry. MCF7 cells were digested by trypsin. Anti-His tag-AF488, anti-EpCAM-APC, and anti-Vimentin-PE antibodies were added.

**Figure 2 ijms-24-12072-f002:**
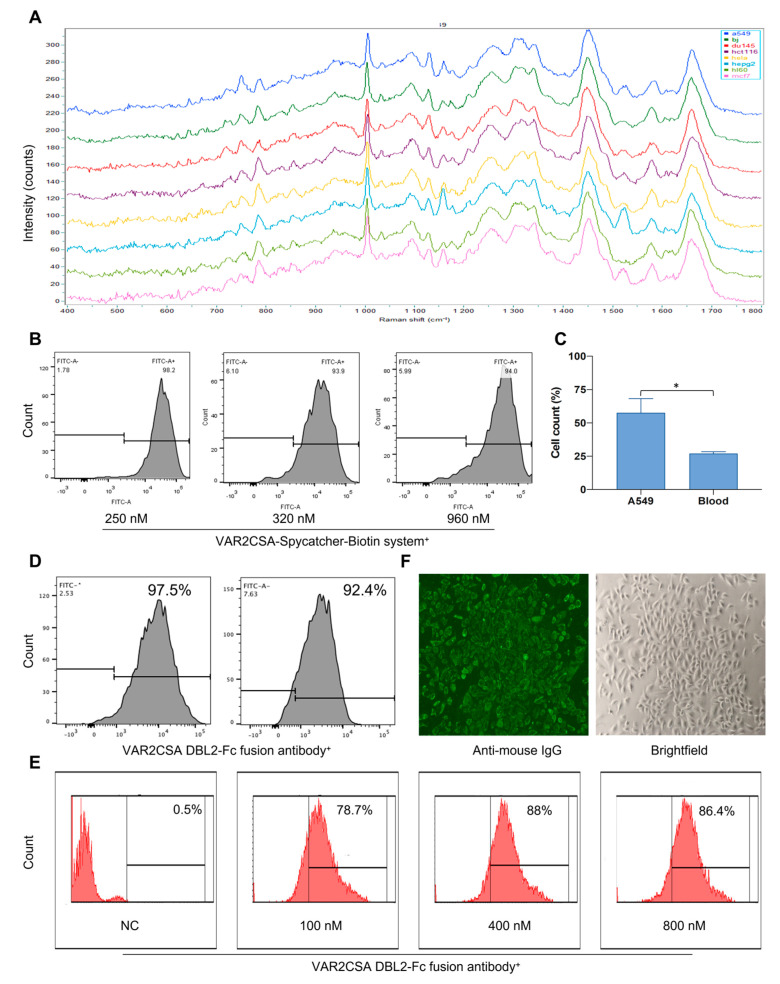
Raman spectroscopy can identify tumor cell types captured by malaria proteins. (**A**) Raman spectra of eight cell lines. Raman spectra were detected after the digestion and fixation of eight different cells. (**B**) Detection of the binding affinity of the VAR2CSA-Spycatcher-Biotin system to A549 cells by flow cytometry. The A549 cells were digested by trypsin. The VAR2CSA-Spycatcher-Biotin system was diluted to the corresponding concentration and incubated with A549 for 30 min at room temperature. The VAR2CSA-Spycatcher-Biotin system (FITC) was labeled with the anti-His Tag-AF488 antibody and incubated at 4 °C for 30 min in the dark. (**C**) Detection of the capture ability of the VAR2CSA-Spycatcher-Biotin system for tumor cells and leukocytes. A549 cells were digested by trypsin. Blood was obtained from mouse hearts after the lysis of erythrocytes. A total of 250 nM of the VAR2CSA-SpyCatcher biotin system was added to the mixed cells, incubated for 2 h at room temperature, and then combined with anti-biotin magnetic beads for 15 min at 4 °C. The cells were captured by an immunomagnetic cell sorter, and the cells were counted before and after the bead capture. * *p* < 0.05 when compared to the leukocyte group. (**D**) Detection of the binding affinity of the VAR2CSA DBL2-Fc fusion antibody to tumor cells by flow cytometry. The tumor cells were digested by trypsin. The VAR2CSA DBL2-Fc fusion antibody was added to the cells and incubated for 30 min at room temperature. Anti-mouse IgGAF488 antibody (FITC) was added and incubated at 4 °C for 2 h in the dark. Left: A549 cells; right: HCT116 cells. (**E**) Detection of the binding affinities of different concentrations of the VAR2CSA DBL2-Fc fusion antibody to A549 by flow cytometry. A549 cells were digested by trypsin. The VAR2CSA DBL2-Fc fusion antibody was diluted to different concentrations, added to the cells, and incubated at room temperature for 30 min. The anti-mouse IgG-AF488 (FITC) antibody was added and incubated at 4 °C for 2 h in the dark. (**F**) Detection of the binding affinity of the VAR2CSA DBL2-Fc fusion antibody to A549 by fluorescence microscopy. A549 cells were grown overnight in 96-well plates. A total of 100 nM of the VAR2CSA DBL2-Fc fusion antibody was added to the cells and incubated for 30 min at 37 °C with 5% CO2. Anti-mouse IgG-AF488 antibody was added and incubated for 2 h at 4 °C in the dark. The cells were photographed under a fluorescence microscope with a 10× objective.

**Figure 3 ijms-24-12072-f003:**
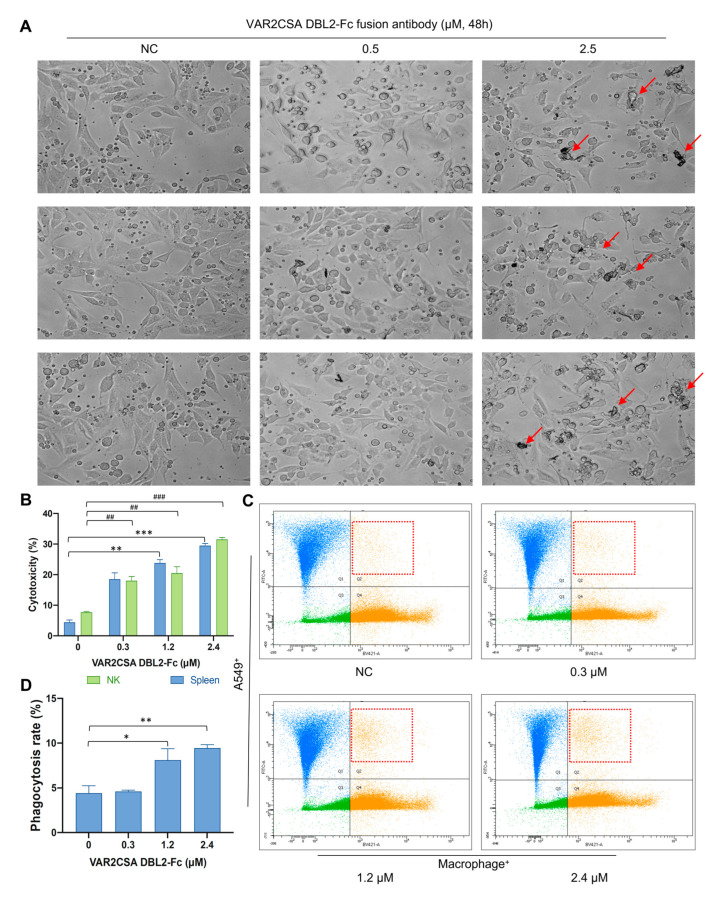
The VAR2CSA DBL2-Fc fusion antibody enhances macrophage antitumor activity and macrophage-dependent phagocytosis in vitro (**A**). Detection of the ADCC effect of VAR2CSA DBL2-Fc fusion antibody on A549 with a microscope. A549 cells were grown in 96-well cells overnight. The VAR2CSA DBL2-Fc fusion antibody was diluted to different concentrations and added to the cells. Mouse spleen cells were added to the 96-well plate at a ratio of 1:5 to tumor cells and cultured for 48 h. The morphology of cells was observed and photographed under a microscope. The red arrow indicted the dead cells. (**B**) Detection of ADCC effect of the VAR2CSA DBL2-Fc fusion antibody on A549 by LDH assay. A549 cells were incubated overnight with different concentrations of fusion antibodies for 2 h. Mouse spleen and NK cells were mixed with A549 cells in a 5:1 ratio and incubated for 6 h. The cell supernatants were used for the LDH assay. ** *p* < 0.01, *** *p* < 0.001 when compared with the vehicle of the spleen group and ^##^ *p* < 0.01, ^###^ *p* < 0.001 when compared with the vehicle of the NK group. (**C**) Detection of the ADCP effect of the VAR2CSA DBL2-Fc fusion antibody on A549 by flow cytometry. A549 cells were mixed with mouse macrophages at a ratio of 2.5:1 and incubated with different concentrations of the fusion antibody for 3 h. Green CellTracker and purple CellTracker were added to label A549 cells (FITC) and macrophages (BV421), respectively. The red boxes are labeled tumor cells being phagocytosed by macrophages. (**D**) The relative quantification of (**C**) was performed. The ADCP effect was calculated as Q2/Q2+Q4. * *p* < 0.05, ** *p* < 0.01 when compared to the vehicle group.

**Table 1 ijms-24-12072-t001:** Relative counts of mouse blood cells and A549.

	Microsphere	CD11b	CD45	Cancer
Plasma	90,014	1352	1155	1911
PBMC	2186	3952	14,784	43,038
Ficoll	19,226	18,306	55,174	2764
Red cell	944	3476	7495	496

**Table 2 ijms-24-12072-t002:** Absolute count of A549 in mouse blood cells.

	a	CD11b/a	CD45/a	Cancer/a
Plamsa	1.072	1261.67 (0.23%)	40,136.55 (0.07%)	1783.32 (0.1%)
PBMC	0.026	151,860.93 (28%)	150,737.11 (38.46%)	1,653,793.23 (96.61%)
Ficoll	0.229	79,980.44 (14.75%)	2,325,415.39 (16.32%)	12,076.15 (0.71%)
Red cell	0.011	309,305.09 (57.02%)	2,777,519.64 (45.15%)	44,135.59 (2.58%)

a: Microsphere/84,000.

**Table 3 ijms-24-12072-t003:** Modeling with the same batch cell sample using six algorithms to predict the cell types.

Tested Cell Type	Tested Cell Type	Predicted Cell Type	KNN Predicted Accuracy	SVM Predicted Accuracy	RF Predicted Accuracy	LDA Predicted Accuracy	PLS-DA Predicted Accuracy	XGB Predicted Accuracy
A549	63	A549	98%	100%	100%	100%	100%	100%
BJ	79	BJ	96%	100%	100%	100%	100%	100%
DU145	71	DU145	100%	100%	100%	100%	92%	100%
HCT116	72	HCT116	97%	100%	100%	100%	99%	100%
Hela	64	Hela	98%	100%	100%	100%	97%	100%
HepG2	78	HepG2	99%	100%	100%	100%	97%	100%
HL60	75	HL60	99%	100%	100%	100%	100%	100%
MCF7	79	MCF7	97%	100%	100%	100%	99%	100%

**Table 4 ijms-24-12072-t004:** Predicting cell types from different batches with the established RF algorithm model.

Test Cell Type	Tested Cell Number	Predicted Accuracy	Predicted Cell Type
A549	62	46.8%	A549
Hela	60	43.3%	Hela
MCF7	62	33.8%	HCT116
HL60	50	60%	HL60
DU145	58	32.7%	BJ

**Table 5 ijms-24-12072-t005:** Predicting laboratory-cultured A549 cell types with multiple classification algorithms.

Algorithm	SVM	K-NN	PLS-DA	RF	LDA	XGB
HCT116	21.05%	38.6%	1.75%	15.79%	8.77%	26.32%
MCF7	7.02%	5.26%	10.53%	7.02%	22.81%	14.04%
A549	21.05%	7.02%	38.6%	14.04%	15.79%	8.77%
Hela	5.26%	0.00%	35.09%	0.00%	14.04%	5.26%
HepG2	24.56%	1.75%	12.28%	12.28%	29.82%	8.77%
DU145	17.54%	38.6%	1.75%	26.32%	7.02%	26.32%
HL60	0.00%	0.00%	0.00%	0.00%	0.00%	0.00%
BJ	3.51%	8.77%	0.00%	24.56%	1.75%	10.53%
Predicted cell type	HepG2	HCT116	A549	DU145	HepG2	HepG2/HCT116

**Table 6 ijms-24-12072-t006:** Predicting malaria-protein-captured A549 cell types with multiple classification algorithms.

Algorithm	SVM	K-NN	PLS-DA	RF	LDA	XGB
HCT116	21.28%	40.43%	6.38%	14.89%	19.15%	27.66%
MCF7	6.38%	2.13%	19.15%	8.51%	42.55%	12.77%
A549	53.19%	6.38%	31.91%	19.15%	21.28%	29.79%
Hela	2.13%	4.26%	27.66%	0.00%	4.26%	0.00%
HepG2	4.26%	0.00%	12.77%	6.38%	8.51%	6.38%
DU145	2.13%	19.15%	2.13%	4.26%	0.00%	2.13%
HL60	0.00%	0.00%	0.00%	0.00%	0.00%	0.00%
BJ	10.64%	27.66%	0.00%	46.81%	4.26%	21.28%
Predicted cell type	A549	HCT116	A549	BJ	MCF7	A549

**Table 7 ijms-24-12072-t007:** Predicting malaria-protein-captured A549 cell types by the laboratory-cultured A549 cell model with six analysis algorithms.

Algorithm	SVM	K-NN	PLS-DA	RF	LDA	XGB
HCT116	4.26%	6.38%	0.00%	4.26%	4.26%	0.00%
MCF7	8.51%	2.13%	10.64%	2.13%	46.81%	2.13%
A549	74.47%	42.55%	34.04%	46.81%	34.04%	89.36%
Hela	0.00%	0.00%	46.81%	0.00%	6.38%	0.00%
HepG2	4.26%	2.13%	8.51%	4.26%	6.38%	4.26%
DU145	0.00%	2.13%	0.00%	0.00%	2.13%	0.00%
HL60	0.00%	0.00%	0.00%	0.00%	0.00%	0.00%
BJ	8.51%	44.68%	0.00%	42.55%	0.00%	4.26%
Predicted cell type	A549	BJ	Hela	A549	MCF7	A549

**Table 8 ijms-24-12072-t008:** Predicting malaria-protein-captured A549 cell types co-cultured with mouse leukocytes by six analysis algorithms.

Algorithm	SVM	K-NN	PLS-DA	RF	LDA	XGB
HCT116	20.00%	20.00%	0.00%	0.00%	40.00%	26.67%
MCF7	0.00%	6.67%	6.67%	0.00%	6.67%	6.67%
A549	0.00%	6.67%	46.67%	0.00%	0.00%	0.00%
Hela	33.33%	13.33%	20.00%	13.33%	6.67%	13.33%
HepG2	0.00%	0.00%	20.00%	6.67%	26.67%	13.33%
DU145	20.00%	46.67%	6.67%	13.33%	6.67%	6.67%
HL60	6.67%	0.00%	0.00%	13.33%	6.67%	13.33%
BJ	20.00%	6.67%	0.00%	53.33%	6.67%	20.00%
Predicted cell type	Hela	DU145	A549	BJ	HCT116	HCT116

## Data Availability

All original data can be requested from the first author.

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
