# Peer review of "Raman Spectroscopy Combined with Malaria Protein for Early Capture and Recognition of Broad-Spectrum Circulating Tumor Cells"

_ijms, 2023, doi:10.3390/ijms241512072_

Round 1

Reviewer 1 Report

The title should indicate what CTCs means. Acronyms should not be used because it is not clear what the article is about.

Before the third paragraph of the introduction (line 62), the authors should better introduce the Raman technique and indicate that it can be used in different areas such as feeding, for surface adsorption, trace analysis , electrochemical detection and biomedical detection, among others.

On the other hand, they must indicate that this technique is simple, fast and non-destructive since this type of advantages make this technique very versatile. In my opinion, they should better indicate and sell the advantages offered by the technique that the authors are using.

The authors should be careful with numbers and units. Sometimes they put it together and other times separately. For example, in line 158 they put 4ºC (together) and 30 min (separate). Please normalize this throughout the entire manuscript.

In my opinion, I think it is a good work and should be published. The application seems really interesting to me and I consider that the authors should continue working in this line since the results are very promising.

English is fine. I suggest just checking a little for small bugs there.

Author Response

Dear reviewer:

Thanks very much for the comments to us and for allowing us to revise the manuscript (ijms-2497250). We really appreciate all your helpful comments and suggestions, which will prove invaluable in revising and improving our paper. Based on the instructions, the file of the revised manuscript was uploaded. Accordingly, we have uploaded a copy of the original manuscript with all the changes highlighted by using the track changes mode in MS Word.

We have carefully studied your suggestion point to point and revised the manuscript accordingly. The amendments are listed as follows:

The title should indicate what CTCs means. Acronyms should not be used because it is not clear what the article is about.

Before the third paragraph of the introduction (line 62), the authors should better introduce the Raman technique and indicate that it can be used in different areas such as feeding, for surface adsorption, trace analysis, electrochemical detection and biomedical detection, among others.

On the other hand, they must indicate that this technique is simple, fast and non-destructive since this type of advantages make this technique very versatile. In my opinion, they should better indicate and sell the advantages offered by the technique that the authors are using.

The authors should be careful with numbers and units. Sometimes they put it together and other times separately. For example, in line 158 they put 4ºC (together) and 30 min (separate). Please normalize this throughout the entire manuscript.

In my opinion, I think it is a good work and should be published. The application seems really interesting to me and I consider that the authors should continue working in this line since the results are very promising.

Based on the reviewers' comments, we have made the following changes:

  1. Thank you for the title suggested. The manuscript title has been revised. Circulating tumor cells have been used instead.
  2. Thank you for the constructive comments. We have stated the application of the Raman technique in different areas and emphasized the advantage of the technique, such as simple, fast, and non-destructive, as suggested by the reviewers (line 65-96).
  3. We completely agree with the reviewer that it is important to use the units properly. We have reviewed the relevant literature, which shows that spaces are required between numbers and English units but not between degrees Celsius and %.
  4. Thank you very much for pointing out the sentence issues in our manuscript. We have careful recheck and had the manuscript polished in writing according to the comments.

Thank you very much for your time and kind consideration.

Sincerely yours,

Xinning Liu

Reviewer 2 Report

The work presented is very interesting for the subject matter and for the different perspectives related to the diagnosis and treatment of oncological diseases. The search for circulating tumor cells (CTC) is of great interest because their identification gives the doctor the possibility to intervene promptly, possibly correct the therapy and therefore act in the most suitable way possible.The use of a microfluidic assay based on Raman spectroscopy for the identification of CTCs would be desirable above all in terms of timeliness of diagnosis and also due to the currently very high costs. Certainly there is still much to be done to improve the specificity of the survey but this work represents a good start. As for the corrections to be made, I would suggest the authors delete lines 35 to 41.

English quality is good, just needs minor corrections.

Author Response

(The authors gave the same response as above.)
